# Task-Based and Resting-State Functional MRI in Observing Eloquent Cerebral Areas Personalized for Epilepsy and Surgical Oncology Patients: A Review of the Current Evidence

**DOI:** 10.3390/jpm13020370

**Published:** 2023-02-19

**Authors:** Hussain Khalid Al-Arfaj, Abdulaziz Mohammad Al-Sharydah, Sari Saleh AlSuhaibani, Soliman Alaqeel, Tarek Yousry

**Affiliations:** 1Medical Imaging Department, King Fahad Specialist Hospital, Dammam 31444, Saudi Arabia; 2Diagnostic and Interventional Radiology Department, King Fahd Hospital of the University, Imam Abdulrahman Bin Faisal University, Dammam 34221, Saudi Arabia; 3Medical Imaging Department, Dammam Medical Complex, Ministry of Health, Dammam 11176, Saudi Arabia; 4Division of Neuroradiology and Neurophysics, Lysholm Department of Neuroradiology, UCL IoN, UCLH, London NW1 2BU, UK

**Keywords:** functional neuroimaging, seizures, neurophysiology, brain tumor, epilepsy surgery

## Abstract

Functional magnetic resonance imaging (fMRI) is among the newest techniques of advanced neuroimaging that offer the opportunity for neuroradiologists, neurophysiologists, neuro-oncologists, and neurosurgeons to pre-operatively plan and manage different types of brain lesions. Furthermore, it plays a fundamental role in the personalized evaluation of patients with brain tumors or patients with an epileptic focus for preoperative planning. While the implementation of task-based fMRI has increased in recent years, the existing resources and evidence related to this technique are limited. We have, therefore, conducted a comprehensive review of the available resources to compile a detailed resource for physicians who specialize in managing patients with brain tumors and seizure disorders. This review contributes to the existing literature because it highlights the lack of studies on fMRI and its precise role and applicability in observing eloquent cerebral areas in surgical oncology and epilepsy patients, which we believe is underreported. Taking these considerations into account would help to better understand the role of this advanced neuroimaging technique and, ultimately, improve patient life expectancy and quality of life.

## 1. Introduction

Functional magnetic resonance imaging (fMRI) is a well-known neuroimaging modality for evaluating patients with brain tumors or who require surgical treatment for epilepsy [1]. It plays a significant role in defining eloquent and other functional brain areas for preoperative planning in cases of intracranial diseases. This technique dynamically enhances anatomical information obtained by conventional MRI. fMRI makes it possible to observe cerebral functions during the performance of specific tasks and in the resting state [2]. Resting-state fMRI can be used to investigate simultaneous activations, which occur in the absence of a task or stimulus between spatially different areas, to determine resting-state networks. Over the past few years, the interest in applying resting-state fMRI and functional connectivity analysis has significantly increased [3]. In this comprehensive review article, we discuss task-based and resting-state fMRI studies and present prospective medical applications in observing eloquent cerebral areas personalized for surgical oncology and epilepsy patients.

## 2. The BOLD Effect

fMRI can be executed using multiple techniques; however, blood oxygen level-dependent functional magnetic resonance imaging (BOLD) is a standard technique commonly used in human brain imaging. The BOLD technique uses blood as an essential disparity material, which replaces the vascular application of paramagnetic disparity materials or radioactive agents [4]. Neuronal activity related to a particular task leads to an almost instant local rise in blood flow to the underlying brain areas to enhance oxygen and glucose levels. This is referred to as neurovascular coupling [1]. The difference in the magnetic properties of deoxygenated and oxygenated hemoglobin in the capillary beds, with an increased oxygenated to deoxygenated hemoglobin ratio during neural activity, is exploited to produce BOLD contrast [2]. Notably, the BOLD signal measured with conventional fMRI can be influenced by medications that adjust neurovascular coupling, for example, by increasing or decreasing baseline cerebral blood flow [5]. Regional hemodynamic alterations transferred through neurovascular coupling are assessed by high spatial precision fMRI, as follows [4]:Increased local cerebral blood volume (rCBV);Increased local cerebral blood flow (rCBF);Relative rise in oxyhemoglobin levels in capillaries and venous blood.

## 3. Techniques for the Acquisition and Processing of BOLD Images

BOLD measurements are obtained using single-shot echo-planar imaging (EPI) sequences, gradient echo (GRE), or spin-echo (SE) [6,7]. Gradient echo sequences have an advantage over SE sequences, as the former obtains greater BOLD signals, particularly from a venous origin [8,9]. A gradient EPI sequence enables very rapid slice imaging by running all phase encodings after an excitation pulse of <90° (77° for 1.5 T and 80° for 3 T scanners). [10] This is achieved by rapid gradient switching to fill the k-space, leading to a set of tiny gradient echoes within the period of a single T2* decay [4]. fMRI with higher field strength (3.0 T and above) results in a higher signal-to-noise ratio (SNR), permitting greater BOLD disparity sensitivity with greater spatial resolution. However, the susceptibility effects of deoxygenated hemoglobin are greater in high-strength fields and may cause more signal changes than anticipated [10]. In addition, compared to conventional MRI, fMRI is more prone to noise because of random variations of the BOLD signal [11].

The analysis of the fMRI signal is complicated by the fact that the measured BOLD signal is composed of both changes induced by neuronal activation as well as non-neural fluctuations. Here, the first is the signal of interest, whereas the second is considered a nuisance signal, such as drift and motion-related artifacts. Inadequately controlling these types of noise may have a significant impact on the subsequent analysis [12].

Therefore, preprocessing steps are mandatory prior to data analysis. These include the recognition and correction of head motion and the elimination of drifts, and they decrease the effect of artifacts and increase the SNR [13].

## 4. Types of fMRI

The two main clinical fMRI categories are task-based and resting-state techniques. The task-based technique is designed to identify neural reactions to experimental circumstances, which can be viewed as brain activity maps. Other purposes of such studies are to identify differences in brain activity patterns between different experimental stimulations, different subject subgroups, and different sessions (e.g., before and after therapy) [2]. On the contrary, functional connectivity studies (resting-state techniques) aim to identify multiple brain areas with similar temporal activity patterns in a resting condition or in the absence of a defined task [2].

### 4.1. Task-Based Methods

There are three major types of task-based fMRI designs: block, event-related, and mixed designs. In a block design, each condition is presented in the form of a repetitive paradigm, such as particular movements that continue for a prolonged period, with the various blocks of different conditions alternating in time. A baseline (i.e., resting) period might be integrated to represent particular brain areas that are typically more involved in baseline situations [2]. Block designs have various advantages, such as the possibility to collapse over various trials to increase the SNR. Block design experiments are naturally adjusted to detect regions of interest (ROIs) for certain tasks [14,15,16,17,18,19]. Despite their advantages, block design experiments have some limitations. Specifically, block designs cannot differentiate between types of trials within a block (e.g., correct vs. failing trials) and may not recognize the types of trials within a trial or between trial events. In addition, brief responses at the start or end of task blocks cannot be properly analyzed [20,21,22,23,24,25,26].

Event-related fMRI design involves the presentation of a single trial rather than various trials together in a block. These trials are individual tests of different conditions randomly submitted, with enough time between tests to separate successive responses. Event-related designs are classified into slow and rapid designs in which the intertrial intervals range from 1 to 2 and 2 to 6 s, respectively [4]. Event-based designs have several benefits over block designs, especially for cognitive tasks. An advantage of event designs is the ability to present stimuli in a random order, minimizing the cognitive fit and subject expectations.

Another crucial benefit of event-related designs is that the response details for various types of tests (and even for individual tests) can be approximated by the event-related average [4]. The disadvantages of event-related designs include a decrease in the SNR, which leads to a lower power output than that of block designs of the same duration [27]. Mixed designs are used to distinguish between short-term and continual long-term activations that persist over the whole block [2]. The trials are displayed in regular and discrete blocks, but there are several event types within each block. Trials rapidly introduced within a block cause short-term dissociable changes in the brain [28]. The modeling of short-term trial activity in mixed designs corresponds to that of event tests in an event-related design. The long-term activation signal is considered a task maintenance signal, which may pertain to the set of tasks [28]. However, poorly designed experiments using mixed designs not only lead to a loss of signal power but also incorrectly distribute signals from one event type to another [28] (Figure 1).

### 4.2. Resting-State Method

Functional connectivity is generally described as “the observed temporal correlation (or other statistical dependencies) between two electro- or neurophysiological measures of different parts of the brain” [29]. A resting network is a group of brain areas with a similarity in their BOLD time sequence acquired while resting [29]. Various resting-state networks have been recognized and named primarily based on the spatial correlation between the resting-state networks and the activation patterns noted in the fMRI studies of a task. The default mode network is the best-known resting-state network. It contains areas of the brain that constantly show a decline in activity when the brain performs any type of cognitive task [29].

Other resting-state networks include the somatomotor network (involving higher-order and primary motor and sensory regions), the visual network, the auditorial network, the linguistic network, the dorsal and ventral attention networks (involved in attentional and cognitive management), and the recently defined cingulo-opercular network, believed to be responsible for tasks demanding executive control [4]. Different analysis methods can be used for resting-state fMRI, including seed analysis, independent component analysis, the graph method, and clustering algorithms. The first and most popular method is seed-based analysis. This method requires selecting ROIs and connecting the mean BOLD time course of voxels within these ROIs and with the time courses of each of the different voxels in the brain. Generally, a threshold is set to identify voxels that significantly correlate with the ROI. However, this approach requires the preselection of an ROI. Following the evaluation of the activation data, the voxel related to the greatest activation is used as a “seed” region to study the resting-state details [30,31,32,33].

Resting-state fMRI can be used to provide information about the intrinsic organization and normal daily functioning of the brain. This may help us to understand how the brain enables complicated data processing and rich sets of ideas, behaviors, and motives. In addition, a better understanding of the brain in its basic resting state may improve the understanding of how it functions in response to the needs of a task. It has good potential to serve as a biomarker for mental disorders, facilitating early diagnosis and allowing assessment of progression [29]. Resting-state fMRI requires only an MRI scanner and is not particularly dependent on patients’ ability to follow commands. As it does not place cognitive demands on the patient, it can possibly be used in people incapable of executing tasks. Therefore, resting-state fMRI can be applied to various age groups from infancy (or even during antenatal development) to older age [29].

## 5. Clinical Applications

Preoperative diagnostic assessment in patients with brain tumors and epilepsy is the most common clinical application of fMRI [4]. The dominant diagnostic goals of presurgical functional neuroimaging are the localization of significant brain areas with respect to the suggested surgical site and the identification of the predominant hemisphere for specific brain operations [4]. A careful preoperative examination of the optimum surgical accessibility and limitations of enucleation in the individual is crucial to avert damage to functionally essential brain structures and to prevent neurological deficits induced by surgery. However, the location and extent of the activated brain sites and the brain activation scheme can vary under pathological conditions; this is known as reorganization or plasticity induced by the lesion [1,4,34].

The advantages of using fMRI over other modalities include the lack of utilizing ionizing radiation, as required in PET and SPECT imaging; being noninvasive, as compared to intraoperative direct electrocortical stimulation (DES); and having better spatial resolution than other noninvasive procedures, such as electromagnetoencephalography [4]. For an optimal examination, fMRI has technical and methodological requirements comprising hardware, software, imaging guidelines, and data evaluation mechanisms. Each patient must promptly repeat the tasks within the scanner under an investigator’s supervision. Overall, all clinical measurements of fMRI must be closely observed by the investigator to cite glitches occurring during paradigm implementation. Clinical fMRI protocols should be enhanced for appropriate clinical examination times, low artifact susceptibility, optimal signal output, and credible localization of efficient brain regions [4].

Studies have shown that 3-T scanners have increased specificity and sensitivity for detecting the localization of motor and somatosensory regions compared to that of 1.5-T scanners [35,36]. Task-based fMRI is more conclusive in low-grade gliomas than in high-grade gliomas because of neurovascular uncoupling, wherein oxygen extraction occurs with no increase in rCBF and rCBV, causing a constant decrease in the fMRI signal despite the increased electrical neural activity. This reduced BOLD sensitivity due to the fact of pathophysiological vascularization in lesional and perilesional regions may lead to faulty unfavorable results [37,38,39].

### 5.1. fMRI of Motor and Somatosensory Functions

The diagnostic objective of fMRI is to locate the primary motor cortex associated with Rolandic lesions since permanent paralysis may be caused by surgery-related injury in the respective functional regions. Two main indications for preoperative fMRI are as follows:When the anatomy is effaced or partially effaced, and morphological Rolandic landmarks cannot be identified due to tumor growth;When a tumor lies in proximity to the identifiable motor hand area.

In brain tumors, changes in representations of different body regions may be detected, besides hemispheric lateralization variations and activation of secondary cortical regions [4]. Somatotopic motor cortex mapping is the primary preoperative fMRI protocol used in patients with Rolandic brain tumors [7]. It consists of three fMRI measurements. Paradigms can include movements of the tongue, fingers, and toes in relation to the tumor to locate the primary motor homunculus potentially affected by the brain tumor [4].

For the optimal clinical setting examination, the paradigms used must be achievable, the motion artifacts must be minimized, and the exam time must be short. In some cases, a short paradigm based on a single body representation of the cortex that provides powerful activation is used. Short acquisition times are crucial in patients experiencing paresis or agitation, as the likelihood of motion artifacts increases with the acquisition period, eventually disturbing the accuracy of the functional localization. In addition, it is possible to minimize the examination time by omitting the smallest pertinent body representations in the case of small lesions that do not seem critical for any motor representation [4].

When defining motor paradigms, it is important to determine if only the main motor cortex should be analyzed or whether secondary areas, such as the supplementary motor area (SMA), should also be included [4]. The SMA is involved in learning, generating action sequences, and the execution of multiple actions involving both sides of the body, and it is located along the medial cerebral cortex in the paracentral lobule and the posterior portion of the superior (medial) frontal gyrus. Alternately, three different stimulating conditions might be integrated into the paradigm: right movement, rest, and left movement. However, in this case, the number of blocks for each paradigm is increased compared to unilateral movements; thus, the examination period and motion sensitivity artifacts are also increased [4].

Movements of the face, arms, legs, and feet have a high probability of motion artifacts, which can degrade the diagnostic value and, thus, need to be minimized [4]. Self-triggered movements are mostly used to assess fMRI activations; however, in some cases, mechanical devices are used for the better evaluation of movements or for measurements of different physical parameters [40]. When a lesion in the primary motor cortex results in paresis, it is reflected by reduced BOLD activation. Therefore, certain appropriate movements to the patient’s situation, including the opposition of finger digits against the thumb, repeated flexing and stretching of the five toes without moving the ankle, and repeated tongue movements with a closed mouth, are needed [4].

Fist clenching/releasing can be attempted in cases of mild paresis of the upper extremities [4]. In addition, intricate finger impedance of the nonparetic hand can be used to formulate ipsilateral robust premotor activation as a further functional landmark to identify the precentral gyrus on the tumor side by localization of the anterior border of the precentral gyrus near the junction of the precentral sulcus with the posterior part of the upper frontal sulcus [4] (Figure 2).

However, in patients with high-grade paresis, the identification of motor function with preoperative fMRI using only auto-triggered movements contralateral to the tumor is difficult. In such patients with motor deficits, passive movements for motor mapping can be used as an alternative method to reliably activate motor brain areas [41]. Some researchers have reported comparable results in active and passive activation tasks. They used a passive palm-finger brushing task in patients who were physically incapable of performing active finger-tapping or hand-squeezing sensorimotor activation tasks [42]. Ideation concerning sequential motor tasks (planning for the motor task without execution) is another technique used to increase cerebral blood flow, which shows some activation in the SMA [43]. Ideation can also show the involvement of the pre-SMA in the early stage of movement planning, which precedes the beginning of voluntary movements [44].

Neurological deficits caused by damage to secondary regions can also occur; however, they are generally transient and not as serious as damage to the main sensory and motor gyri [34]. According to a literature review, fMRI findings were helpful in the single or multiple surgical decisions category in 89% of all examined patients with tumors [45]. High sensitivity and specificity were reported for fMRI localization of the motor region, ranging from 71% to 100% and 68% to 100%, respectively [46,47,48].

In another study in which 25 patients were examined for primary brain tumors in regions close to sensorimotor areas, effective fMRI measurements were acquired in 80% of patients, in 75% of whom the measurements were used for preoperative planning. When the distance between the tumor margin and BOLD activation was ≥10 mm, the possibility for postoperative functional loss was markedly reduced [49]. Another study of 54 patients found that when the distance between the lesion and BOLD activation was <5 mm, there was a high probability of postoperative functional deficits. Thus, cortical stimulation of 10 mm is recommended [50]. A comparison of preoperative fMRI results with intraoperative mapping using DES allowed validation of the fMRI findings, demonstrating that the preoperative fMRI showed high agreement with the intraoperatively obtained data [51] (Figure 3).

Somatosensory somatotopic mapping allows for the evaluation of neuroplastic changes in cortical motor activation. In preoperative fMRI, somatosensory somatotopic mapping is primarily used as a diagnostic complement in cases where motor paradigms are hard to operate [4]. If reduced BOLD activation is caused by inadequate residual function in the motor cortex in cases of paresis, then activation of the primary somatosensory cortex in the postcentral gyrus with lip, finger, and toe representations, which do not necessitate patient cooperation, can be investigated. Somatosensory stimulation can be performed by activating the contralateral primary and bilateral secondary areas [52,53]. Most published studies used nonstandard stimuli, such as manual hand touch; however, some studies relied on automated, reproducible stimulation methods, including electric, tactile, and vibrotactile techniques [4].

### 5.2. fMRI of Language Functions

The diagnostic objectives of fMRI preoperative language function include the localization of language areas relative to brain tumors or epileptic areas, including Broca’s and Wernicke’s areas, and the identification of the dominant hemisphere for language.

fMRI for language-related brain activation is indicated in the following scenarios [4]:Patients with tumor-related linguistic impairments, including right hemisphere tumors, in whom nonstandard organization of relevant cortical representations of the language must be presumed;Patients with no language deficit but with left hemisphere tumors that are anatomically near the inferior frontal gyrus (Broca’s area), superior temporal gyrus (Wernicke’s area), the anterior insula (Dronkers’ area), and the supramarginal or angular gyri (Geschwind area);Left-handed patients, including those with right hemispheric tumors.

The primary purpose of identifying and interpreting complex language networks is to anticipate and reduce postsurgical language deficits [4]. Although multiple language areas have been identified, it is unclear which of them, if resected, would lead to language deficits [4,54,55] (Figure 4).

Language functions are examined using different paradigms involving auditory or visual stimulation [2]. The most common forms of tasks that are efficiently used to assess the lateralization of language in patients with a brain tumor [4] are word-generation tasks (also referred to as verbal eloquence tasks) pursued by semantic decision-making tasks. The former tends to exhibit a fairly persistent activation of the frontal gyri of the leading language hemisphere, and the latter is most effective in activating both the inferior frontal and superior temporal gyri [56]. In word-generation tasks, patients or subjects are presented with a noun of a certain semantic type (e.g., bird and nutrition) and are requested to recover a phonological or semantic-related term [4].

However, in verb-generation tasks, patients are asked to provide a verb in reaction to hearing or seeing a noun [4]. These tasks accurately stimulate the dominant inferior and dorsolateral frontal lobes, including the prefrontal and premotor regions [4]. In the semantic decision-making task, patients are asked to perform mental categorization of abstract and concrete nouns (sentence fulfillment, verb to noun verb generation, and antonymous or synonymous decision task). Motion artifacts could be reduced through nonvocalized language tasks [4].

Unlike the easy-to-conduct movement tasks for motor fMRI, the assessment of language function requires more patient cooperation. Thus, all patients should be properly prepared for fMRI tasks. Intensive training before the examination is the most effective way to ensure satisfactory patient cooperation during clinical fMRI acquisition. This training should ensure the best possible correspondence between the standard fMRI paradigms used and the patient’s linguistic ability to ensure robust functional localization and BOLD signals. Dependable functional localization of the Broca and Wernicke regions can be accomplished within about 4 min per paradigm [4].

The American Society of Functional Neuroradiology supports the recommendation of using at least two paradigms for preoperative language lateralization [57]. Similarly, most European centers use one or more paradigms for determining language lateralization in patients with epilepsy [58]. This is understandable because language is a complex function that consists of five major domains: hearing, reading, speaking, writing, and comprehension. It would, therefore, be difficult to develop a unique and robust paradigm that simultaneously activates these multiple language components [58].

The language areas have large anatomical variations and lack unequivocal cortical landmarks. Therefore, selecting appropriate patients for presurgical fMRI for language mapping is largely independent of their anatomical relationship with specific cortical structures [4]. In patients with epilepsy for whom surgery is being considered, fMRI is generally used to assess the lateralization of language function and, to a minor degree, for determining the intrahemispheric distribution of the eloquent cortex [4]. As mentioned earlier, semantic decision-making tasks are the most effective paradigms to activate both the inferior frontal and superior temporal gyri. However, some of these paradigms require cognitive skills often impaired in patients with epilepsy [57,59]. A preliminary study found that the BOLD signal acquired during basic language and motor function tests was comparable between normal controls and patients with epilepsy, confirming the feasibility of using the technique in people with epilepsy [60]. However, we should consider the inter- and intrahemispheric reorganization of language areas, which can occur in patients with epilepsy according to the location of the pathology [61,62]. Atypical linguistic dominance in patients with epilepsy is associated with the early onset of seizures and low right-hand dominance. The slightly higher prevalence of atypical language portrayal among these patients emphasizes the value of evaluating hemispheric dominance before surgical procedures in areas likely relevant for language in both cerebral hemispheres [4,63,64] (Figure 5).

Word-generation tasks are often used to evaluate pediatric and adult patients with epilepsy being considered for surgery and demonstrate unanimous agreement with the WADA test (which involves intra-arterial catheter-directed infusion of sodium amobarbital or propofol selectively into one hemisphere in awake patients) and DES [65,66,67]. However, some data indicate that young children demonstrate more widespread activity than adults, particularly in verbal fluency tasks [68].

The WADA test is invasive, and it exhibits marked risks, and the accuracy of its solitary outcomes may be weakened by severe drug effects, which can lead to behavioral disorders associated with sedation and agitation. Although the WADA test is typically considered a reference procedure in language lateralization tests, it is not a standard procedure [69]. fMRI is a valuable alternative to the WADA test for assessing hemispheric dominance for language and has substituted the WADA test for presurgical determination of hemispheric language dominance in a few centers [70,71,72]. It has fairly good sensitivity (83.5%) and specificity (88.1%), as measured against the WADA test [73]. fMRI shows more details than the WADA test, which can characterize hemispheric dominance (left vs. right or mixed dominance).

However, in bilateral or an atypical linguistic representation or when fMRI is not definitive for other reasons, additional details regarding linguistic representation require more invasive techniques [63]. Unlike the WADA test, fMRI has the potential to provide detailed maps of the intrahemispheric location of critical language areas, along with information on lateralization. On the contrary, only the WADA test can simulate if a particular function will be retained after a specific part of the hemisphere is resected. Therefore, the WADA test and fMRI do not deliver the same information and are integral [4]. Some authors found Broca’s area activation in 77% of patients and Wernicke’s area in 91% using a mixture of silent word-generation tasks upon hearing spoken words. In addition, they found that the lateralization of language with fMRI and the WADA test was consistent among the 13 patients reviewed [74]. Several intraoperative validation studies showed the good reliability of fMRI in locating the Broca and Wernicke zones and in identifying the dominant language hemisphere. Bizzi et al. [47] found that fMRI had a sensitivity of 80% and a specificity of 78% for language mapping compared to cortical DES [46,47].

In recent reports, the sensitivities and specificities varied from 59% to 100% and from 0% to 97%, respectively [75,76]. The high degree of variability is likely due to the differences among the studies, such as the linguistic paradigms, different magnetic field strengths, and intra-operative methods used, along with the type of lesion and grade of tumor [77,78]. Some regions triggered during language tasks perform a secondary supporting part in language function, and resection of these regions might not necessarily result in a clinically significant shortfall. Thus, at this point, fMRI may be considered useful in facilitating DES but still cannot substitute it [54,55].

### 5.3. fMRI of Visual Functions

Generally, the primary clinical application of fMRI vision mapping is preoperative planning in patients with a pathological vision abnormality or who potentially require visual pathway surgery [79]. fMRI retinotopic mapping offers detailed information on how the visual field matches the cortical representation in the patient. This mapping can be accomplished effectively by the sequential display of a slowly expanding checkerboard ring and a slowly rotating checkerboard wedge.

It is carried out by presenting checkerboards throughout the active period and a blank screen during the resting period [2]. The checkerboard designs consist of high-contrast, white and black checks that counterphase flicker at 8 Hz, leading to powerful neural activation and ultimately large increases in the BOLD signal in visually responsive brain regions. The stimuli are introduced in a temporal phase mapping pattern in which the areas in the visual field varying in eccentricity or angular position are stimulated at separate times. The patient may be asked to stare at a small marker in the middle of the video screen and press a button every time the marker disappears at random. The button task is helpful because it independently checks that the patient visually watches the display throughout the fMRI scan and records the patient’s focus on the stimulus [79].

Eccentric and angular mapping details can be used to create a single functional field map (FFMap). Every circle symbol in the FFMap matches a visually responsive voxel in the cortex, and its location is determined by the eccentricity and angle of the annular and wedge stimuli to which the voxel reacted most strongly. The color of each circle shows the intensity of the fMRI response, while the size represents an overall estimate of the error of the preferred stimulus location. For a healthy individual, the FFMap will contain circular symbols spread over the whole visual field. However, if the visual cortex is focally affected, symbols will be missing or reduced in size throughout the retinotopic zone of the visual field [79] (Figure 6).

A study showed a high agreement between fMRI and DES for visual cortex localization. The primary visual cortex was identified using fMRI in all trial cases [80]. In a separate study, the visual cortex was also identified in all cases using fMRI. The fMRI signal was recognized in all six patients with lesions in the visual cortex reported in this study and was comparable to the signal recognition in healthy volunteers [81].

Ocular dominance consists of a tendency to favor visual input coming from one eye in which subjects’ vision is more precise, and images show up more precise, much more stabilized, and probably larger [82]. Some researchers can determine ocular dominance using fMRI with a high success rate (87–97%) using a visual paradigm [83]. Similarly, the patient was asked to stare at the gray dot in the center while the checkerboard wedge rotated (B) to visualize brain activity in an upside down manner.

## 6. Limitations of Task-Based fMRI

Defining fMRI-based safe resection margins is challenging due to the fact of issues such as different scanners, acquisition paradigms, and postprocessing steps. In addition, based on fMRI, cerebral reorganization patterns in which the location of the brain structures change during surgery and postsurgical deficits, including permanent and transitory deficits, are difficult to predict. Task training is always conducted immediately before fMRI and lasts for a minimum of 20 min. As previously stated, task-based fMRI is difficult to apply if patient cooperation is poor. When patients are subjected to excessive testing, they can only do some of the required tasks, reducing activation. This is also true for the subtest, where few triggers are provided [4].

In addition, “free-thinking” may lead to uncontrolled activation, for example, when patients are bored. For motion artifacts, the movements of the tongue and toes, and the spots on the opposite fingers, are less critical than the movements of the hands, feet, or lips. In addition, nonvocalized language paradigms are slightly more crucial than vocalized tasks. Furthermore, BOLD signals are weaker for tactile stimulation, mainly for those presented to the lower limbs [4].

## 7. Resting-State fMRI Applications

Many publications have explored the use of resting-state fMRI in preoperative planning, primarily for patients with tumors and those requiring surgery for epilepsy [84]. Resting-state fMRI is very less demanding and may be applied to patients who cannot cooperate in task-based paradigms, such as patients with an altered mental state, including patients with Alzheimer’s disease, young children, sedated patients, and patients with aphasia or paresis [84,85,86]. Another advantage of resting-state-based fMRI over task-based fMRI is the ability to identify multiple networks simultaneously, saving scanning time if multiple networks have to be studied [3]. A case report mentions the use of this technique to localize the motor cortex of a patient with a brain tumor [84].

Another study showed the successful localization of the sensorimotor areas by employing seed-based methods for patients with neoplasms or epileptic foci near sensorimotor areas. They found agreement between the resting-state fMRI, task-based fMRI data, and DES [85]. By using resting-state fMRI data, researchers were able to distinguish between subjects with epilepsy of the medial temporal lobe and healthy controls. Based on data from 16 patients with intractable temporal lobe epilepsy and 52 healthy controls, they had an average sensitivity of 77.2% and specificity of 83.86% [87]. A separate study showed higher specificity of resting-state compared to task-based fMRI and was more consistent with DES results [88]. Identifying language regions is more challenging, given that the localization of language areas, which might also be distorted by tumors, is difficult [89]. Moreover, multiple studies have shown that the visual cortex can be selectively mapped using resting-state fMRI [90,91,92,93,94]. At the time of writing, the validity of resting-state fMRI for clinical applications was not fully established, but studies on resting-state fMRI showed promising results. However, further research is required prior to the routine use of resting-state MRI in the clinic; it is necessary to compare the various analysis methods and their efficiency in detecting various pathological conditions in both groups, particularly in individual subjects.

## 8. Up-to-Date Research Endeavors for Task-Based fMRI in Epilepsy and Oncology Patients

There is growing evidence to support task-based fMRI’s ability to locate eloquent areas in relation to various brain lesions. It facilitates preoperative planning by determining language dominance and locating language-receptive areas. However, fMRI is not considered a routine study, probably due to the need for standard protocols and consistent techniques. This makes it difficult to compare studies and affects—largely—the chosen stimulation paradigm [4,95]. Till then, the authors trust that the interpretation of fMRI shall be merely by trained experts in this field as a prerequisite to avoid misinterpreting this promising modality with potential patient harm and induction of research bias [4,95,96,97].

Recent work by Agarwal et al. (2021) [98] investigated the different hemodynamic responses toward performed stimuli and their alteration effect on the BOLD signal. This has proven to be reduced in functional areas of tumor extension or may represent the so-called “steal phenomenon” for functional areas. In addition, when encountering a hypervascular lesion, so-called “neurovascular uncoupling” is a hot research topic [98]. Another era of research is the advent of real-time fMRI that facilitates the rapid analysis of fMRI data, which is potentially beneficial in preprocedural diagnosis by replacing off-line processing, which usually consumes much time [99].

The question remains: How to determine the safe margins or how reliable is fMRI in identifying eloquent areas and their associated risks of functional deficits? [100] Hitherto, fMRI has been considered a helpful tool to facilitate intraoperative electrocortical stimulation. However, more integrity evidence is required to stand on its own. As such, efforts to conduct validation studies in a standardized way are encouraged to address these unanswered questions [4,97,100].

A great example is the recent efforts by the European Society of Neuroradiology-Diagnostic and Interventional (ESNR) Epilepsy Working Group (2020) [59], which evaluated fMRI implementation and interpretation across Europe. They explored language lateralization among epilepsy and brain tumor patients. They showed the superiority of fMRI over the WADA test in terms of cost and risk with enhanced localization capability [59]. Thakkar et al. (2022) [101] looked at whether crossed cerebrocerebellar language lateralization visualizes semantic and phonological tasks. They have proved to be valuable tools for preprocedural planning and have a prognostic value, and lastly, they benefit patients suffering from brain tumors in eloquent regions [101].

## 9. Up-to-Date Research Endeavors for Resting-State fMRI in Epilepsy and Oncology Patients

In recent years, increasing attention has been paid to the role of the development of deep learning in fMRI at rest and its recognition capabilities for language networks [102]. The implementation of deep learning for electroencephalogram (EEG)-fMRI analysis has also captured momentum, because it allows for rapid and sensitive data analysis within a fraction of a time compared to conventional morphometric surface-based methods and at no cost to the robustness [102,103,104].

Additionally, researchers in epilepsy and deep learning have studied rodent models for the role of EEG-fMRI analysis and BOLD signal alteration in epilepsy with generalized spike discharges. They proved that it does not manifest suddenly, as their EEG would suggest, which is a fact that was known, but they are the result of network changes that precede them by approximately 1 min, but this needs to be recognized in human studies [105]. The concept of deep learning in a resting state fMRI has provided future guidance to the available AI technologies that help filter artifacts and analyze functional connectivity in EEG-fMRI, in addition to assessing the etiological dependencies of signal changes in cortical and subcortical networks [106].

Cui et al. (2022) [107] developed a volumetric functional network parcellation approach for resting-state fMRI performed in brain tumor patients to assess their de novo brain tumors before the procedure. They identified different connectivity patterns in 20 brain tumor patients. Their approach can hold a potential clinical utility for presurgical planning for better procedural outcomes. However, other larger studies with fewer confounding factors are still required [107]. Moreover, Wang et al. (2022) [108] performed a pilot study to correlate resting fMRI with cognitive measurements in diffuse glioma survivors by exploring functional connectivity and neuropsychological assessments. They suggest that resting fMRI alterations in functional connectivity are useful surrogates for cognitive and postprocedural morbidity; however, longitudinal methodologies with larger samples are needed [108].

Concerning longitudinal postsurgical outcomes, the current literature lacks a standardized timeline to evaluate the recovery period. Nevertheless, it is not clear that a single time point can reliably reflect optimal recovery. Therefore, for the idealistic tracking of postprocedure recovery, the authors advocate a standardized consensus for optimal scanning times individualized based on case specificities and obtaining multiple acquisitions of several temporal points where possible [109]. Nowadays, future research for personalized patient assessment is of great interest to resting fMRI. Conducting robust, personalized analyses would enable clinicians to have accurate diagnostic and prognostic capabilities for patient-specific care [109,110].

## 10. Conclusions

Task-based fMRI has shown its validity and high sensitivity in the localization of various representations of the human body in the primary motor cortex, the localization of language areas, and the function of lateralized language before surgical procedures in patients with brain tumors or epilepsy. In contrast, resting-state fMRI has offered new insights into the functional architecture of the healthy brain. While clinical applications of resting-state fMRI are still limited, numerous prospective clinical applications are presently being investigated, including presurgical planning for patients with brain tumor or epilepsy. Since it is noninvasive and does not require patient cooperation, resting-state fMRI might be specifically useful in patients who cannot be examined using current methods of functional and lesion localization.

## Figures and Tables

**Figure 1 jpm-13-00370-f001:**
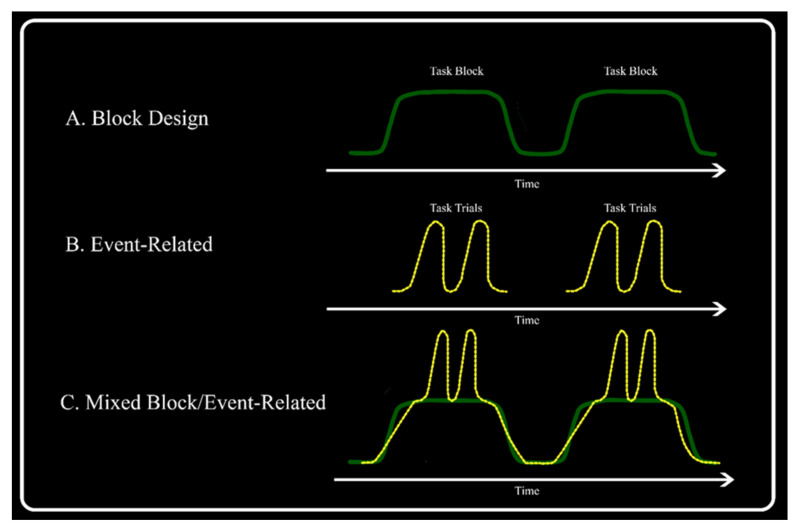
Contrasting experimental fMRI designs distinguish between various tasks-related signals: (**A**) the solid, green line represents the block design, wherein the task block represents a prolonged period of a repetitive paradigm separated by rest; (**B**) the dashed, yellow line represents the event-related task trials that exhibit different conditions presented in a random sequence; (**C**) mixed-type design with transient activities (trials) presented throughout the sustained activity (task block).

**Figure 2 jpm-13-00370-f002:**
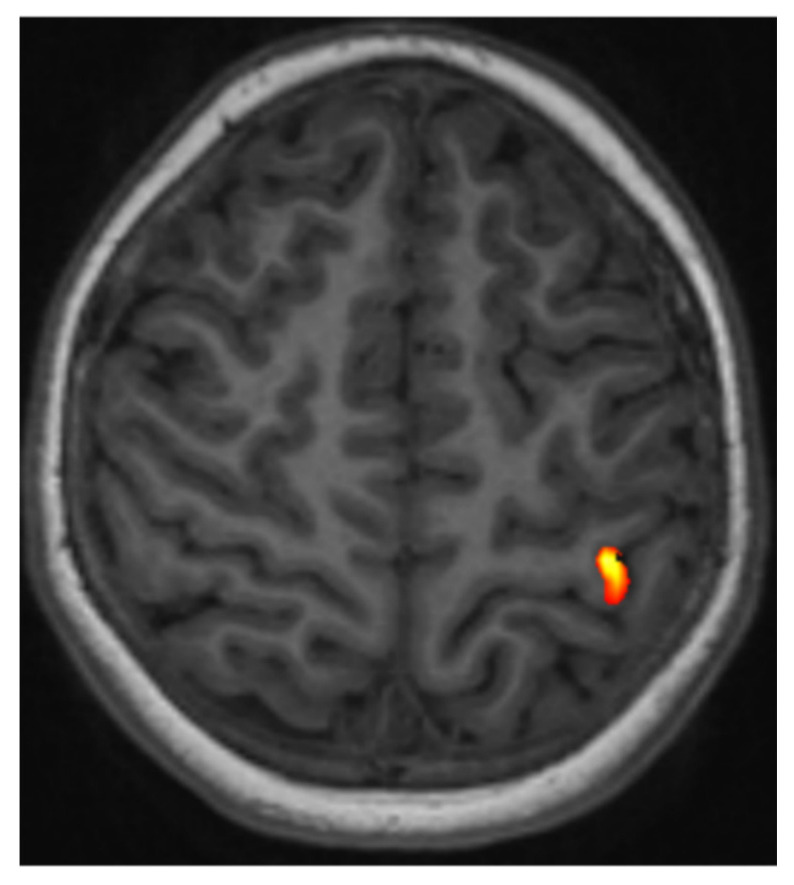
An exemplary fMRI of a healthy volunteer showing an intraparenchymal area of activation in the precentral gyrus (hand knob area).

**Figure 3 jpm-13-00370-f003:**
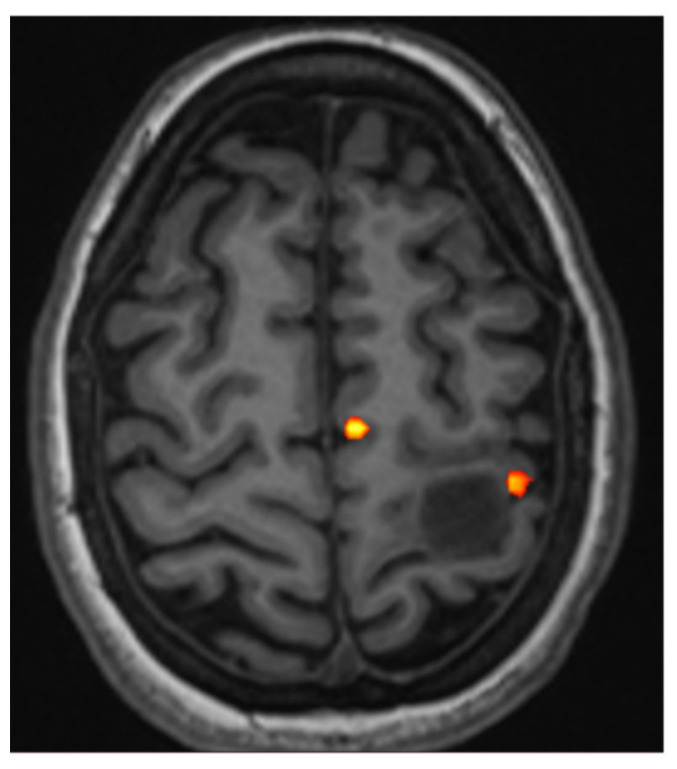
An exemplary fMRI of a patient with a left frontal lobe mass at the precentral gyrus (hand knob area) showing lateral displacement of the parenchymal associated with hand activation. Additional activation in the medial cerebral cortex of the left paracentral sulcus just anterior to the left paracentral lobule represents the supplementary motor area (SMA).

**Figure 4 jpm-13-00370-f004:**
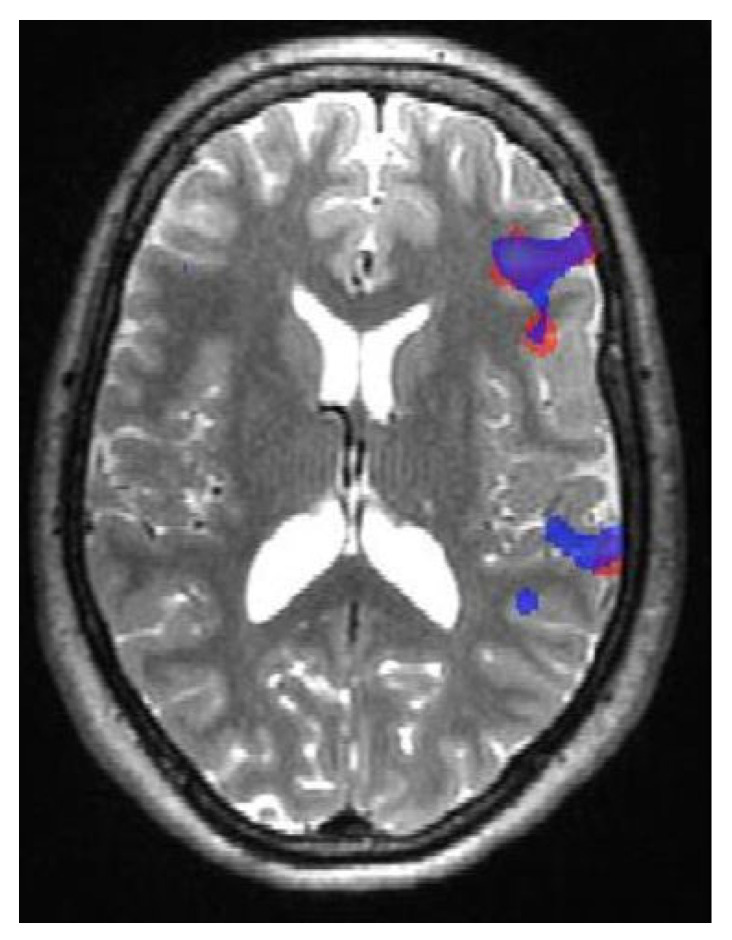
Functional localization of Broca’s and Wernicke’s areas using verbal fluency (red) and verb generation (blue) paradigms in a control patient.

**Figure 5 jpm-13-00370-f005:**
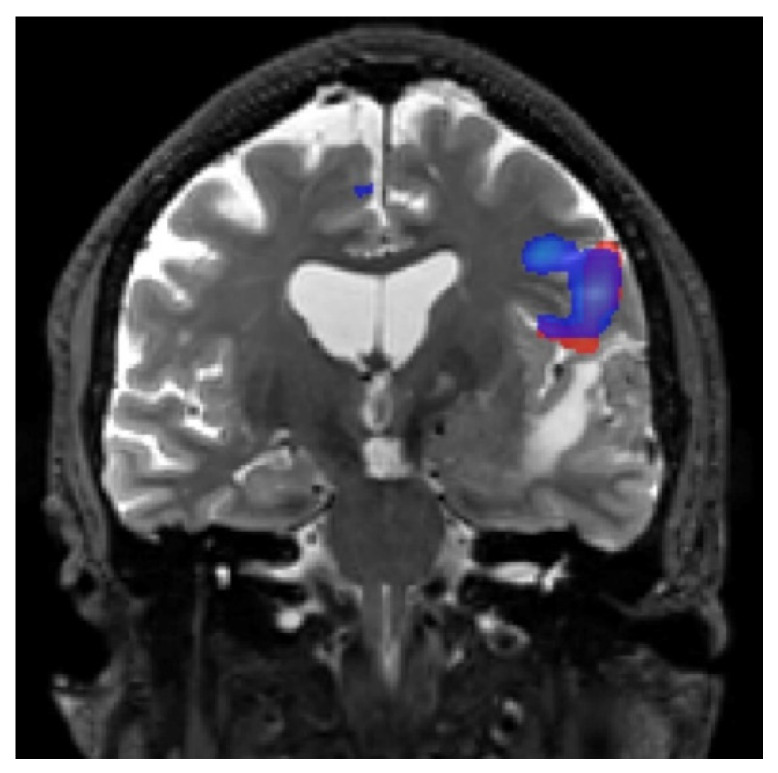
fMRI of activation in Broca’s area as an indicator of a left hemispheric language dominance area using verbal fluency (red) and verb generation (blue) paradigms in a patient with left temporal lobe mass. In addition, there is a superior displacement of the parenchymal area associated with language activation.

**Figure 6 jpm-13-00370-f006:**
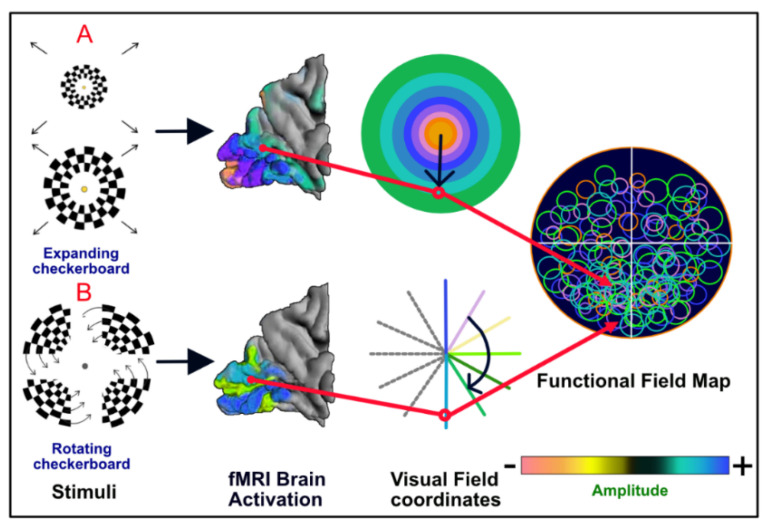
Stimuli for visual field mapping and construction of functional field maps (FFMaps). Retinotopic mapping stimuli consist of an expanding black and white checkerboard ring (**A**) or a rotating checkerboard wedge (**B**). The patient was asked to stare at the yellow dot in the center while the rings of flickering stimuli were outside of his/her visual field (**A**) to separate the signal from the fovea from the periphery of the visual field.

## Data Availability

Not applicable because this is a review of publicly available information.

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
