# Peer review of "Task-Based and Resting-State Functional MRI in Observing Eloquent Cerebral Areas Personalized for Epilepsy and Surgical Oncology Patients: A Review of the Current Evidence"

_jpm, 2023, doi:10.3390/jpm13020370_

Round 1

Reviewer 1 Report

This is a clinically oriented fMRI review. It provides a good overview about use of different kind of fMRI in observing eloquent cerebral areas in epilepsy and oncology patients before surgery. It is clearly written and covers relevant literature. I have only a few minor comments about the technical part.

Line 74: “…after 90 deg pulse”. In gradient echo EPI flip angels also less than 90 deg are commonly used, when shorter TR is needed (typically at or slightyy below Ernest angel)

 Line 74-75: “by turning the gradients off and on” This may give a bit wrong impression as for example read gradient goes contantly uo and down …Maybe something like  “ by rapid gradient swithing”” would work better.

Line 80-81: “, fMRI is more prone to noise because of random variations of the BOLD signal “ This is a bit unclear sentence that might need rephrasing . Random variation in signal is noise. If it is BOLD signal in the sense that it becomes from physiological sources (blood oxygenation, CBF, CBV) then variation is what you want to measure…

Author Response

Reviewer 1

This is a clinically oriented fMRI review. It provides a good overview about use of different kind of fMRI in observing eloquent cerebral areas in epilepsy and oncology patients before surgery. It is clearly written and covers relevant literature. I have only a few minor comments about the technical part.

Response: Thank you for spotting light on the rationale behind our study. We hope that the improvements made in this version are appropriate.

Line 74: “…after 90 deg pulse. In gradient echo EPI flip angels also less than 90 deg are commonly used, when shorter TR is needed (typically at or slightyy below Ernest angel)

Response: Thank you for this comment. We have revised this part as follows:

Gradient EPI sequence enables very rapid slice imaging by running all phase encodings after an excitation pulse of < 90° (77° for 1.5T and 80° for 3T scanners). [10]

[10] Filippi. fMRI techniques and protocols. Ed. Massimo Filippi. Vol. 830. Humana press, 2016

 Line 74-75: by turning the gradients off and on This may give a bit wrong impression as for example read gradient goes contantly uo and down Maybe something like  by rapid gradient swithing”” would work better.

Response: Thank you for this comment. We have revised this part as follows:

This is achieved by rapid gradient switching to fill the k-space, leading to a set of tiny gradient echoes within the period of a single T2* decay [4].’

[4] Stippich C, editor Clinical functional MRI; Presurgical functional neuroimaging: Presurgical functional neuroimaging. Springer; 2015.

Line 80-81: , fMRI is more prone to noise because of random variations of the BOLD signal This is a bit unclear sentence that might need rephrasing . Random variation in signal is noise. If it is BOLD signal in the sense that it becomes from physiological sources (blood oxygenation, CBF, CBV) then variation is what you want to measure

Response: Thank you. We do agree that the raised point needs further clarification. Therefore, we have clarified and revised this part as follows:

‘The analysis of the fMRI signal is complicated by the fact that the measured BOLD signal is composed of both changes induced by neuronal activation, as well as non-neural fluctuations. Here, the first is the signal of interest, whereas the second is considered a nuisance signal like drift, and motion-related artifacts. Not adequately controlling these types of noise may have a significant impact on the subsequent analysis [12].’

[12] Lindquist, Martin A., et al. "Modular preprocessing pipelines can reintroduce artifacts into fMRI data." Human brain mapping 40.8 (2019): 2358-2376.

  • Response: We have incorporated the changes suggested by the first reviewer in the revised manuscript. We hope that we have adequately addressed all the issues raised and that our manuscript is now suitable for publication in Journal of Personalized Medicine.

Reviewer 2 Report

The authors presented a narrative review of current applications of fMRI technologies in clinical and scientific scenarios in neuroscience. The review is well written and a general overview of techniques, purposes and pitfalls are presented to the reader with a sufficient body of literature as reference list. Despite the aim of the manuscript is appropriate and might find attention in the journal's readers, I judge the level of focus of the study quite preliminary and general, being less likely to be of interest to mid-beginners and experts in the field of MRI-base neuroscience. 

In particular, the study does not provide but an introduction of these techniques, being partially critical on pros and limitations as widely reported in the discussion section, not sufficient to researchers approaching this field, eventually looking for literature reference to get involved in research experiments. 
The previous literature comprehends a sufficient number of articles reviewing the basics concepts of fMRI techniques in brain tumours and epilepsy-affected patients. 
For this reason, I consider this manuscript not suitable for publication in its current form.  

Author Response

Reviewer 2

The authors presented a narrative review of current applications of fMRI technologies in clinical and scientific scenarios in neuroscience. The review is well written and a general overview of techniques, purposes and pitfalls are presented to the reader with a sufficient body of literature as reference list.

Response: Thank you for spotting light on the rationale behind our study. We hope that the improvements that have been made in this version are appropriate.

Despite the aim of the manuscript is appropriate and might find attention in the journal's readers, I judge the level of focus of the study quite preliminary and general, being less likely to be of interest to mid-beginners and experts in the field of MRI-base neuroscience.

Response: Thank you. We do agree with the respectable reviewer that we intended to make this review a general primer for practitioners dealing with epilepsy and/ or brain tumor patients. However, at your request in this round, we amend our manuscript to include new parts that would be of great interest to mid-beginners and experts such as discussing up-to-date advancements, research endeavors, and few of unanswered questions. We hope that the improvements been made in this version are appropriate.

In particular, the study does not provide but an introduction of these techniques, being partially critical on pros and limitations as widely reported in the discussion section, not sufficient to researchers approaching this field, eventually looking for literature reference to get involved in research experiments.

Response: The authors would like to thank the respectable reviewer for spotlighting a crucial aspect that needs to include in our review. We agree that adjoining pieces of writing to attract researchers and readers who are more specialized in fMRI is paramount. Therefore, in this round, we included sections to address this point and ensured to add only updated resources with evidence-based integrity as follows:

‘8. Up-to-date research endeavors for task-based fMRI in epilepsy and oncology patients

There is growing evidence to support task-based fMRI's ability to locate elequent areas in relation to various brain lesions. It facilitates preoperative planning by determining language dominance and locating language-receptive areas. However, fMRI is not considered a routine study, probably due to the need for standard protocols and consistent techniques. This makes it difficult to compare studies and affects - largely - the chosen stimulation paradigm [95, 4]. Till then, the authors trust that the interpretation of fMRI shall be merely by trained experts in this field as a prerequisite to avoid misinterpreting this promising modality with potential patient harm and induction of research bias [4, 95-97].

Recent work by Agarwal et al. (2021) [98] investigated different hemodynamic responses toward performed stimuli and their alteration effect on the BOLD signal. This has proven to be reduced in functional areas of tumor extension or may represent the so-called "steal phenomenon" for functional areas. Also, when encountering a hypervascular lesion, so-called “neurovascular uncoupling” is a hot research topic [98]. Another era of research is the advent of real-time fMRI facilitates rapid analysis of fMRI data, potentially beneficial in preprocedural diagnosis by replacing off-line processing, which usually consumes much time [99].

The question remains: How to determine the safe margins or how reliable is fMRI in identifying eloquent areas and their associated risks of functional deficits? [100] Hitherto, fMRI has been considered a helpful tool to facilitate intraoperative electrocortical stimulation. But more integrity evidence is required to stand on its own. As such, efforts to conduct validation studies in a standardized way are encouraged to address these unanswered questions [4, 97, 100].

A great example is the recent efforts by the European Society of Neuroradiology-Diagnostic and Interventional (ESNR) Epilepsy Working Group (2020) [59], which evaluated fMRI implementation and interpretation across Europe. They explored language lateralization among epilepsy and brain tumor patients. They showed the superiority of fMRI over the Wada test in terms of cost and risk with enhanced localization capability [59]. Thakkar et al. (2022) [101] looked at whether crossed cerebrocerebellar language lateralization visualizes semantic and phonological tasks. They have proved valuable tools for pre-procedural planning and have a prognostic value. Lastly, for the benefit of patients suffering from brain tumors in eloquent regions [101]

  1. Up-to-date research endeavors for resting-state fMRI in epilepsy and oncology patients

In recent years, more and more attention has been paid to the role of the development of deep learning in fMRI at rest and its recognition capabilities for language networks [102]. The implementation of deep learning for Electroencephalogram (EEG)-fMRI analysis has also captured momentum because it allows rapid and sensitive data analysis within a fraction of a time compared to conventional morphometric surface-based methods and at no cost to the robustness [102-104].

 Additionally, researchers in epilepsy and deep learning have studied rodent models for the role of EEG-fMRI analysis and BOLD signal alteration in epilepsy with generalized spike discharges. They proved that it does not manifest suddenly, as their EEG would suggest, which is a fact that was known, but they are the result of network changes that precede them by around 1 min, but this needs to be recognized in human studies [105]. The concept of deep learning in a resting state fMRI has provided future guidance to available AI technologies that help filter artifacts and analyze functional connectivity in EEG-fMRI. In addition, assess the etiological dependencies of signal changes in cortical and subcortical networks [106].

Cui et al. (2022) [107] developed a volumetric functional network parcellation approach for resting-state fMRI performed in brain tumor patients; to assess their de novo brain tumors before the procedure. They identified different connectivity patterns in 20 brain tumor patients. Their approach can hold a potential clinical utility for presurgical planning for better procedural outcomes. However, other larger studies with fewer confounding factors are still required [107]. Besides, Wang et al. (2022) [108] performed a pilot study to correlate resting fMRI with cognitive measurements in diffuse glioma survivors by exploring functional connectivity and neuropsychological assessments. They suggest that resting fMRI alterations in functional connectivity are useful surrogates for cognitive and postprocedural morbidity; however, longitudinal methodologies with larger samples are needed [108].

Concerning longitudinal postsurgical outcomes, the current literature lacks a standardized timeline to evaluate the recovery period. Nevertheless, it is not clear that a single time point can reliably reflect optimal recovery. Therefore, for idealistic tracking for post-procedure recovery, the authors advocate standardized consensus for optimal scanning times individualized based on case specificities and obtaining multiple acquisitions of several temporal points- where possible [109]. Nowadays, future research for personalized patient assessment is of great interest to resting fMRI. Conducting robust, personalized analyses would enable clinicians to have accurate diagnostic and prognostic capabilities for patient-specific care [109, 110].’

The previous literature comprehends a sufficient number of articles reviewing the basics concepts of fMRI techniques in brain tumours and epilepsy-affected patients. For this reason, I consider this manuscript not suitable for publication in its current form. 

Response: Thank you. We hope that the improvements made in this version are appropriate.

  • Response: We have incorporated the changes suggested by the second reviewer in the revised manuscript. We hope that we have adequately addressed all the issues raised and that our manuscript is now suitable for publication in the Journal of Personalized Medicine.
